# Implicit Regularization Effects of Unbiased Random Label Noises with SGD

## Abstract

Random label noises (or observational noises) widely exist in practical machine learning settings. we analyze the learning dynamics of stochastic gradient descent (SGD) over the quadratic loss with unbiased label noises, and investigate a new noise term of dynamics, which is dynamized and influenced by mini-batch sampling and random label noises, as an implicit regularizer. Our theoretical analysis finds such implicit regularizer would favor some convergence points that could stabilize model outputs against perturbation of parameters. To validate our analysis, we use our theorems to analyze the implicit regularizer of SGD with unbiased random label noises for linear regression via Ordinary Least-Square (OLS), where the numerical simulation backups our theoretical findings. We further extend our proposals to interpret the newly-fashioned noisy self-distillation tricks for deep learning, where the implicit regularizer demonstrates a unique capacity of selecting models with improved output stability through learning from well-trained teachers with additive unbiased random label noises.

## 1 Introduction

Stochastic Gradient Descent (SGD) has been widely used as an effective way to train deep neural networks with large datasets (Bottou, 1991). While the mini-batch sampling strategy was firstly proposed to lower the cost of computation per iteration, it has been consider to incorporate an implicit regularizer preventing the learning process from converging to the local minima with poor generalization performance (Zhang et al., 2017; Zhu et al., 2019; Jastrzebski et al., 2017; Hoffer et al., 2017; Keskar et al., 2017). To interpret such implicit regularization, one can model SGD as gradient descent (GD) with gradient noises caused by mini-batch sampling (Bottou et al., 2018). Studies have demonstrated the potentials of such implicit regularization or gradient noises to improve the generalization performance of learning from both theoretical (Mandt et al., 2017; Chaudhari & Soatto, 2018; Hu et al., 2019b; Simsekli et al., 2019) and empirical aspects (Zhu et al., 2019; Hoffer et al., 2017; Keskar et al., 2017). In summary, gradient noises keep SGD away from converging to the sharp local minima that generalizes poorly (Zhu et al., 2019; Hu et al., 2019b; Simsekli et al., 2019) and would select a flat minima (Hochreiter & Schmidhuber, 1997) as the outcome of learning.

In this work, we aim at investigating the influence of random label noises to the implicit regularization under mini-batch sampling of SGD. To simplify our research, we assume the training dataset as a set of vectors $\mathcal{D} = \{x_1, x_2, x_3, \ldots, x_N\}$. The label $\tilde{y}_i$ for every vector $x_i \in \mathcal{D}$ is the noisy response of the true neural network $f^*(x)$ such that

$$\tilde{y}_i = y_i + \varepsilon_i, \ y_i = f^*(x_i), \text{ and } \mathbb{E}[\varepsilon_i] = 0, \ \text{var}[\varepsilon_i] = \sigma^2 \ , \tag{1}$$

where the label noise $\varepsilon_i$ is assumed to be an independent zero-mean **random variable**. In our work, the random label noises can be either (1) drawn from probability distributions before training steps (but dynamized by mini-batch sampling of SGD) or (2) realized by the random variables per training iteration (Han et al., 2018). Thus learning is to approximate $f(x, \widehat{\theta})$ that beats $f^*(x)$, such that

$$\widehat{\theta} \leftarrow \underset{\forall \theta \in \mathbb{R}^d}{\operatorname{argmin}} \left\{ \frac{1}{N} \sum_{i=1}^{N} \tilde{L}_i(\theta) := \frac{1}{N} \sum_{i=1}^{N} (f(x_i, \theta) - \tilde{y}_i)^2 \right\}. \tag{2}$$

Inspired by (Hochreiter & Schmidhuber, 1997; Zhu et al., 2019), our work studies how unbiased label noises $\varepsilon_i$ $(1 \leq i \leq N)$ would affect the "selection" of $\widehat{\theta}$ from possible solutions, in the viewpoint of learning dynamics (Saxe et al., 2014) of SGD under mini-batch sampling (Li et al., 2017; Wu et al., 2020; Hu et al., 2019b).

**Contributions.** Our analysis shows that under mild conditions, with gradients of label-noisy losses, SGD might incorporate an additional data-dependent noise term, complementing with the stochastic gradient noises (Li et al., 2017; Wu et al., 2020) of label-noiseless losses, through re-sampling the samples with label noises (Li et al., 2018) or dynamically adding noises to labels over iterations (Han et al., 2018). We consider such noises as an implicit regularization caused by unbiased label noises, and interpret the effects of such noises as a solution selector of learning procedure. More specifically, this work has made unique contributions as follow.

**(1) Implicit Regularizer.** We reviewed the preliminaries (Li et al., 2017; Ali et al., 2019; Hu et al., 2019b; Wu et al., 2020) and extent the analytical framework in (Li et al., 2017) to interpret the effects of unbiased label noises as an additional implicit regularizer on top of the continuous-time dynamics of SGD. Through discretizing the continuous-time dynamics of label-noisy SGD, we write discrete-time approximation to the learning dynamics, denoted as $\theta_k^{\mathrm{ULN}}$ for $k = 1, 2, \ldots$, as

$$\theta_{k+1}^{\mathrm{ULN}} \leftarrow \theta_k^{\mathrm{ULN}} - \frac{\eta}{N} \sum_{i=1}^{N} \nabla L_i^*(\theta_k^{\mathrm{ULN}}) + \xi_k^*(\theta_k^{\mathrm{ULN}}) + \xi_k^{\mathrm{ULN}}(\theta_k^{\mathrm{ULN}}), \tag{3}$$

where $L_i^*(\theta) = (f(x_i, \theta) - f^*(x_i))^2$ refers to the *label-noiseless loss function* with sample $x_i$ and the true (noiseless) label $y_i$, the noise term $\xi_k^*(\theta)$ refers to the stochastic gradient noise (Li et al., 2017) of label-noiseless loss function $L_i^*(\theta)$, then we can obtain the new implicit regularizer caused by the unbiased label noises (ULN) for $\forall \theta \in \mathbb{R}^d$, which can be approximated as follow

$$\xi_k^{\mathrm{ULN}}(\theta) \approx \sqrt{\frac{\eta}{B}} \left( \frac{\sigma^2}{N} \sum_{i=1}^{N} \nabla_\theta f(x_i, \theta) \nabla_\theta f(x_i, \theta)^\top \right)^{\frac{1}{2}} z_k, \text{ and } z_k \sim \mathcal{N}(\mathbf{0}_d, \mathbf{I}_d), \tag{4}$$

where $z_k$ refers to a random noise vector drawn from the standard Gaussian distribution, $\theta_k$ refers to the parameters of network in the $k^{th}$ iteration, $(\cdot)^{1/2}$ refers to the Chelosky decomposition of the matrix, $\nabla_\theta f(x_i, \theta) = \partial f(x_i, \theta)/\partial \theta$ refers to the gradient of the neural network output for sample $x_i$ over the parameter $\theta_k$, and $B$ and $\eta$ are defined as the learning rate and the batch size of SGD respectively. Obviously, the strength of such implicit regularizer is controlled by $\sigma^2$, $B$ and $\eta$.

**(2) Effects to Linear Regression.** To understand behaviors of the implicit regularizer $\xi_t^{\mathrm{ULN}}(\theta_t)$ to the learning dynamics, we studied SGD over Ordinary Least Square (OLS). With the proposed model, we could easily obtain the implicit regularizer such that

$$\xi_k^{\mathrm{ULN}}(\beta) \approx \sqrt{\eta/B} \left( \sigma^2 \bar{\Sigma}_N \right)^{1/2} z_k \text{ and } z_k \sim \mathcal{N}(\mathbf{0}_d, \mathbf{I}_d) \tag{5}$$

where $\bar{\Sigma}_N = \frac{1}{N} \sum_{i=1}^{N} x_i x_i^\top$ referring to the sample covariance matrix of the training dataset. Our theoretical elaboration suggests that SGD with unbiased random label noises would converge to a distribution of Gaussian-alike centered at the optimal solution of OLS, where the span and shape of the distribution would controlled by $\sigma^2$ and $\bar{\Sigma}_N$ when $\eta$ and $B$ are constant. We conducted the extensive experiments using SGD with various $\sigma^2$ and $\bar{\Sigma}_N$, and successfully obtain the results that coincide our theories and directly visualize the effects of the implicit regularizer over the path of learning and converging distribution of SGD for noisy linear regression.

**(3) Inference Stabilizer.** The regularization effects of unbiased random label noises should be

$$\mathbb{E}_{z_k} \left\| \xi_k^{\mathrm{ULN}}(\theta_k) \right\|_2^2 \approx \frac{\eta \sigma^2}{BN} \sum_{i=1}^{N} \|\nabla_\theta f(x_i, \theta_k)\|_2^2 = \frac{\eta \sigma^2}{BN} \sum_{i=1}^{N} \left\| \frac{\partial}{\partial \theta} f(x_i, \theta_k) \right\|_2^2, \tag{6}$$

where $\nabla_\theta f(x, \theta)$ refers to the gradient of $f$ over $\theta$ and the effects is controlled by the batch size $B$ and the variance of label noises $\sigma^2$. We extend above results to understand the newly-fashioned noisy self-distillation (Zhang et al., 2019a; Kim et al., 2020) paradigms, where a well-trained model is supposed to be further improved through learning from its noisy outputs. Our analysis showed that, when the new convergence achieved, the noisy self-distillation strategy would prefer to re-select a model with a lower neural network gradient norm $\frac{1}{N} \sum_{i=1}^{N} \|\nabla_\theta f(x_i, \theta)\|_2^2$, where the gradient norm characterizes the variation/instability of neural network inference results (over perturbations) around parameters of the model. We carry out extensive experiments while results backup our theories. Note that while the earlier work (Bishop, 1995) found training with input noises can also bring regularization effects, our work focuses on the observational noises on labels.

## 2 PRELIMINARIES AND RELATED WORK

**SGD Dynamics and Implicit Regularization**    We follow settings in (Li et al., 2017) and consider SGD as an algorithm that, in the $k^{th}$ iteration with the estimate $\theta_k$, it randomly picks up a $B$-length subset of samples from the training dataset i.e., $B_k \subset \mathcal{D}$, and estimates the mini-batch stochastic gradient $\frac{1}{b} \sum_{\forall x_i \in B_k} \nabla L_i(\theta_k)$, then updates the estimate for $\theta_{k+1}$ based on $\theta_k$, as follow

$$\theta_{k+1} \leftarrow \left( \theta_k - \frac{\eta}{B} \sum_{\forall x_i \in B_k} \nabla L_i(\theta_k) \right) = \left( \theta_k - \frac{\eta}{N} \sum_{\forall x_i \in \mathcal{D}} \nabla L_i(\theta_k) + \sqrt{\eta} V_k(\theta_k) \right), \quad (7)$$

where $\eta$ refers to the step-size of SGD, and $V_k(\theta_k)$ refers to a stochastic gradient noise term caused by mini-batch sampling. The noise would converge to zero with increasing batch size, as follow

$$V_k(\theta_k) = \sqrt{\eta} \left( \frac{1}{N} \sum_{\forall x_i \in \mathcal{D}} \nabla L_i(\theta_k) - \frac{1}{B} \sum_{\forall x_i \in B_k} \nabla L_i(\theta_k) \right) \to \mathbf{0}_d, \text{ as } B \to N. \quad (8)$$

Let define $\Sigma_N^{\text{SGD}}(\theta_k)$ as the sample covariance matrix of loss gradients $\nabla L_i(\theta_k)$ for $1 \leq i \leq N$, where we follow (Li et al., 2017) and do not make low-rank assumptions on $\Sigma_N^{\text{SGD}}(\theta_k)$. Under mild conditions (Li et al., 2017; Chaudhari & Soatto, 2018), one can approximate SGD as $\bar{\theta}_k$ such that

$$\bar{\theta}_{k+1} \leftarrow \bar{\theta}_k - \frac{\eta}{N} \sum_{\forall x_i \in \mathcal{D}} \nabla L_i(\bar{\theta}_k) + \xi_k(\bar{\theta}_k), \ \xi_k(\bar{\theta}_k) = \sqrt{\frac{\eta}{B}} \left( \Sigma_N^{\text{SGD}}(\bar{\theta}_k) \right)^{\frac{1}{2}} z_k, \ z_k \sim \mathcal{N}(0, \mathbf{I}_d). \ (9)$$

The implicit regularizer of SGD could be considered as $\xi_k(\bar{\theta}_k) = \sqrt{\eta/B} \left( \Sigma_N^{\text{SGD}}(\bar{\theta}_k) \right)^{\frac{1}{2}} z_k$ which is data-dependent and controlled by the learning rate $\eta$ and batch size $B$ (Smith et al., 2018). (Mandt et al., 2017; Chaudhari & Soatto, 2018; Hu et al., 2019b) discussed SGD for varational inference and enabled novel applications to samplers (Zhang et al., 2019b; Xiong et al., 2019). To understand the effect to generalization performance, (Zhu et al., 2019; Smith et al., 2018) studied the escaping behavior from the sharp local minima (Keskar et al., 2017) and convergence to the flat ones. Finally, (Gidel et al., 2019) studied regularization effects to linear DNNs and (Wu et al., 2020) proposed new multiplicative noises to interpret SGD and obtain stronger theoretical properties.

**SGD Implicit Regularization for Ordinary Least Square (OLS)**    The most recent and relevant work in this area is (Ali et al., 2019; 2020), where the same group of authors studied the implicit regularization of gradient descent and stochastic gradient descent for OLS. They investigated an implicit regularizer of $\ell_2$-norm alike on the parameter, which regularizes OLS as a Ridge estimator with decaying penalty. Prior to these efforts, F. Bach and his group have studied the convergence of gradient-based solutions for linear regression with OLS and regularized estimators under both noisy and noiseless settings in (Dieuleveut et al., 2017; Marteau-Ferey et al., 2019; Berthier et al., 2020).

**Self-Distillation and Noisy Students**    Self-distillation (Zhang et al., 2019a; Xie et al., 2020; Xu et al., 2020; Kim et al., 2020) has been examined as an effective way to further improve the generalization performance of well-trained models. Such strategies enable knowledge distillation using the well-trained ones as teacher models and optionally adding noises (e.g., dropout, stochastic depth, and label smoothing or potentially the label noises) onto training procedure of student models.

**Discussion on the Relevant Work**    Though tremendous pioneering studies have been done in this area, we still make contributions in above three categories. First of all, this work characterizes the implicit regularization effects of label noises to SGD dynamics. Compared to (Ali et al., 2019; 2020) working on linear regression, our model interpreted general learning tasks. Even from linear regression perspectives (Ali et al., 2019; 2020; Berthier et al., 2020), we precisely measured the gaps between SGD dynamics with and without label noises using the continuous-time diffusion.Compared to (Lopez-Paz et al., 2016; Kim et al., 2020), our analysis emphasized role of the implicit regularizer caused by label noises for model selection, where models with high inferential stability would be selected. (Li et al., 2020) is the most relevant work to us, where authors studied the early stopping of gradient descent under label noises via neural tangent kernel (NTK) (Jacot et al., 2018) approximation. Our work made the analyze for SGD without assumptions for approximation such as NTK. To best of our knowledge, this work is the first to understand the effects of unbiased label noises to SGD dynamics, by addressing technical issues including implicit regularization, OLS, self-distillation, model selection, and the stability inference results.

## 3 Learning Dynamics and Implicit Regularization of SGD with Unbiased Random Label Noises

From initialization $\theta_0^{\text{ULN}}$, SGD with Unbiased Random Label Noises uses an iterative algorithm that updates the estimate incrementally. Specifically, in the $k^{th}$ iteration, SGD randomly picks up a batch of sample $B_k \subseteq \mathcal{D}$ to estimate the stochastic gradient, as follow

$$\tilde{g}_k(\theta_k^{\text{ULN}}) = \frac{1}{|B_k|} \sum_{x_i \in B_k} \nabla \tilde{L}_i(\theta_k^{\text{ULN}}) = \frac{1}{N} \sum_{i=1}^{N} \nabla L_i^*(\theta_k^{\text{ULN}}) + \xi_k^*(\theta_k^{\text{ULN}}) + \xi_k^{\text{ULN}}(\theta_k^{\text{ULN}}), \quad (10)$$

where $\nabla L_i^*(\theta)$ for $\forall \theta \in \mathbb{R}^d$ refers to the loss gradient based on the label-noiseless sample $(x_i, y_i)$ and $y_i = f^*(x_i)$, $\xi_k^*(\theta)$ refers to stochastic gradient noises (Li et al., 2017) through mini-batch sampling over the gradients of label-noiseless samples, and $\xi_k^{\text{ULN}}(\theta)$ is an additional noise term caused by the mini-batch sampling and the Unbiased Random Label Noises, such that

$$\nabla L_i^*(\theta) = \frac{\partial}{\partial \theta}(f(x_i, \theta) - f^*(x_i))^2 = (f(x_i, \theta) - f^*(x_i)) \cdot \nabla f(x_i, \theta),$$

$$\xi_k^*(\theta) = \frac{1}{|B_k|} \sum_{x_j \in B_k} \left( \nabla L_j^*(\theta) - \frac{1}{N} \sum_{i=1}^{N} L_i^*(\theta) \right), \text{ and } \mathbb{E}_{B_k}[\xi_k^*(\theta)] = \mathbf{0}_d, \quad (11)$$

$$\xi_k^{\text{ULN}}(\theta) = -\frac{1}{|B_k|} \sum_{x_j \in B_k} \varepsilon_j \cdot \nabla f(x_j, \theta), \text{ and } \mathbb{E}_{B_k, \varepsilon_i}[\xi_k^{\text{ULN}}(\theta)] = \mathbf{0}_d.$$

Note that, for every iteration $\forall \theta \in \mathbb{R}^d$, the random vectors $\xi_k^*(\theta)$ and $\xi_k^{\text{ULN}}(\theta)$ are with zero-mean as $\mathbb{E}(\varepsilon_j) = 0$. To characterize the variances of the two random vectors, we define two matrix-value functions $\Sigma_N^{\text{SGD}}(\theta)$ and $\Sigma_N^{\text{ULN}}(\theta)$ over $\theta \in \mathbb{R}^d$ based on the label-noiseless losses, such that

$$\Sigma_N^{\text{SGD}}(\theta) = \frac{1}{N} \sum_{j=1}^{N} \left( \nabla L_j^*(\theta) - \frac{1}{N} \sum_{i=1}^{N} L_i^*(\theta) \right) \left( \nabla L_j^*(\theta) - \frac{1}{N} \sum_{i=1}^{N} L_i^*(\theta) \right)^{\top}$$

$$\Sigma_N^{\text{ULN}}(\theta) = \frac{\sigma^2}{N} \sum_{j=1}^{N} \nabla_\theta f(x_j, \theta) \nabla_\theta f(x_j, \theta)^{\top} \text{ as } \text{var}[\varepsilon_j] = \sigma^2. \quad (12)$$

Under mild conditions, we have $\text{var}[\xi_k^*(\theta)] = 1/B \cdot \Sigma_N^{\text{SGD}}(\theta)$ and $\text{var}[\xi_k^{\text{ULN}}(\theta)] = 1/B \cdot \Sigma_N^{\text{ULN}}(\theta)$.

**SGD Learning Dynamics with Unbiased Random Label Noises** We consider the SGD algorithm with unbiased random label noises in the form of gradient descent with additive data-dependent noise. When $\eta \to 0$, we assume the noise terms $\xi_k^*(\theta_k)$ and $\xi_k^{\text{ULN}}(\theta_k)$ are independent, then we can follow the analysis in (Hu et al., 2019a) to derive the diffusion process of SGD with unbiased random label noises, denoted as $\theta^{\text{ULN}}(t)$ over continuous-time $t \geq 0$, such that

$$\mathbf{d}\theta^{\text{ULN}} = -\frac{1}{N} \sum_{i=1}^{N} \nabla L_i(\theta^{\text{ULN}})\mathbf{d}t + \sqrt{\frac{\eta}{B}} \left( \Sigma_N^{\text{SGD}}(\theta^{\text{ULN}}) \right)^{\frac{1}{2}} \mathbf{d}W_1(t)$$

$$+ \sqrt{\frac{\eta}{B}} \left( \Sigma_N^{\text{ULN}}(\theta^{\text{ULN}}) \right)^{\frac{1}{2}} \mathbf{d}W_2(t), \quad (13)$$

where $W_1(t)$ and $W_2(t)$ refer to two independent Brownie motions over time and $\mathbf{d}t = \sqrt{\eta}$. Again, we can obtain the discrete-time approximation (Li et al., 2017; Chaudhari & Soatto, 2018) to the SGD dynamics, denoted as $\bar{\theta}_k^{\text{ULN}}$ for $k = 1, 2, \ldots$, which in the $k^{th}$ iteration behaves as

$$\bar{\theta}_{k+1}^{\text{ULN}} \leftarrow \bar{\theta}_k^{\text{ULN}} - \frac{\eta}{N} \sum_{i=1}^{N} \nabla L^*(\bar{\theta}_k^{\text{ULN}}) + \sqrt{\frac{\eta}{B}} \left( \left( \Sigma_N^{\text{SGD}}(\bar{\theta}_k^{\text{ULN}}) \right)^{\frac{1}{2}} z_k + \left( \Sigma_N^{\text{ULN}}(\bar{\theta}_k^{\text{ULN}}) \right)^{\frac{1}{2}} z_k' \right), \quad (14)$$

where $z_k$ and $z_k'$ are two independent $d$-dimensional random vectors drawn from a standard $d$-dimensional Gaussian distribution $\mathcal{N}(\mathbf{0}_d, \mathbf{I}_d)$ per iteration independently, and $\theta_0^{\text{ULN}} = \theta^{\text{ULN}}(t = 0)$. Note that the errors from the SGD algorithm to its continuous-time diffusion process and from the continuous-time dynamics to its discretization are bounded under weak convergence (Hu et al., 2019a). In this way, we can use the trajectory of the discrete-time dynamics $\bar{\theta}_k^{\text{ULN}}$ to analyze the behaviors of SGD algorithm $\theta_k^{\text{ULN}}$ over iterations.

**Implicit Regularizer Influenced by Unbiased Random Label Noises**  Compared the stochastic gradient with unbiased random label noises $\tilde{g}_k(\theta)$ and the stochastic gradient based on the label-noiseless losses, we find an additional noise term $\xi_k^{\mathrm{ULN}}(\theta)$ as the **implicit regularizer**.

To interpret $\xi_k^{\mathrm{ULN}}(\theta)$, we first define the diffusion process of SGD based on Label-NoiseLess losses i.e., $L_i^*(\theta)$ for $1 \le i \le N$ as $\mathbf{d}\theta^{\mathrm{LNL}} = -\frac{1}{N}\sum_{i=1}^N \nabla L_i^*(\theta^{\mathrm{LNL}})\mathbf{d}t + \sqrt{\frac{\eta}{B}}\left(\Sigma_N^{\mathrm{SGD}}(\theta^{\mathrm{LNL}})\right)^{\frac{1}{2}}\mathbf{W}(t)$. Through comparing $\theta^{\mathrm{ULN}}(t)$ with $\theta^{\mathrm{LNL}}(t)$, the effects of $\xi_k^{\mathrm{ULN}}(\theta)$ over continuous-time form should be $\sqrt{\eta/B}(\Sigma_N^{\mathrm{ULN}}(\theta))^{1/2}\mathbf{d}W(t)$. Then, in discrete-time, we could get results as follow.

**Proposition 1 (The implicit regularizer $\xi_k^{\mathrm{ULN}}(\theta)$)** *The implicit regularizer of SGD with unbiased random label noises could be approximated (with $O(\sqrt{\eta})$ approximation error due to discretization (Li et al., 2017)) as follow,*

$$\xi_k^{\mathrm{ULN}}(\theta) \approx \sqrt{\frac{\eta}{B}}\left(\frac{\sigma^2}{N}\sum_{i=1}^N \nabla_\theta f(x_i,\theta)\nabla_\theta f(x_i,\theta)^\top\right)^{\frac{1}{2}} z_k, \text{ and } z_k \sim \mathcal{N}(\mathbf{0}_d, \mathbf{I}_d). \quad (15)$$

*In this way, we can estimate the expected strength of the implicit regularizer $\xi_k^{\mathrm{ULN}}(\theta)$ as follow,*

$$\mathbb{E}_{z_k}\|\xi_k^{\mathrm{ULN}}(\theta)\|_2^2 \quad = \frac{\eta\sigma^2}{BN}\sum_{i=1}^N \|\nabla_\theta f(x_i,\theta)\|_2^2 . \quad (16)$$

*In this way, we can conclude that the effects of implicit regularization caused by unbiased random label noises for SGD is proportional to $\frac{1}{N}\sum_{i=1}^N \|\nabla_\theta f(x_i,\theta)\|_2^2$ – the average gradient norm of the neural network $f(x,\theta)$ over samples. Please refer to appendix for the proof.*

**Inference Stabilizer**  Here we extend the existing results on SGD (Zhu et al., 2019; Wu et al., 2018) to understand **Proposition 1** as follows.

(1) Inference Stability -  The gradient norm $\frac{1}{N}\sum_{i=1}^N \|\nabla_\theta f(x_i,\theta)\|_2^2 = \frac{1}{N}\sum_{i=1}^N \|\frac{\partial}{\partial\theta}f(x_i,\theta)\|_2^2$ characterizes the variation of neural network output $f(x,\theta)$ based on samples $x_i$ (for $1 \le i \le N$) over the parameter interpolation around the point $\theta$. Lower $\frac{1}{N}\sum_{i=1}^N \|\nabla_\theta f(x_i,\theta)\|_2^2$ comes higher stability of neural network $f(x,\theta)$ outputs against the (random) perturbations over parameters.

(2) Escape and Converge -  When the noise $\xi_k^{\mathrm{ULN}}(\theta)$ is $\theta$-dependent (section 4 would present a special case that $\xi_k^{\mathrm{ULN}}(\theta)$ is $\theta$-independent with OLS), we follow (Zhu et al., 2019) and suggest that the implicit regularizer help SGD escape from the point $\tilde{\theta}$ with high neural network gradient norm $\frac{1}{N}\sum_{i=1}^N \|\nabla_\theta f(x_i,\tilde{\theta})\|_2^2$, as the scale of noise $\xi_k^{\mathrm{ULN}}(\tilde{\theta})$ is large. Reciprocally, we follow (Wu et al., 2018) and suggest that when the SGD with unbiased random label noises converges, the converging point $\theta^*$ should be with small $\frac{1}{N}\sum_{i=1}^N \|\nabla_\theta f(x_i,\theta^*)\|_2^2$.

(3) Performance Tuning -  Considering $\eta\sigma^2/B$ as the coefficient balancing the implicit regularizer and vanilla SGD, one can regularize/penalize the SGD learning procedure with the fixed $\eta$ and $B$ more fiercely using a larger $\sigma^2$. More specifically, we could expect to obtain solutions with lower $\frac{1}{N}\sum_{i=1}^N \|\nabla_\theta f(x_i,\theta)\|_2^2$ or higher inference stability of neural networks, as regularization effects become stronger when $\sigma^2$ increases.

## 4 IMPLICIT REGULARIZATION EFFECTS TO LINEAR REGRESSION

Here, we consider a special example of SGD with unbiased random label noises using linear regression, where a simple quadratic loss function is considered for OLS, such that

$$\widehat{\beta}_{\mathrm{OLS}} \leftarrow \underset{\beta \in \mathbb{R}^d}{\mathrm{argmin}} \left\{\frac{1}{N}\sum_{i=1}^N \tilde{L}_i(\beta) := \frac{1}{N}\sum_{i=1}^N \left(x_i^\top\beta - \tilde{y}_i\right)^2\right\}, \quad (17)$$

where samples are generated through $\tilde{y}_i = x_i^\top\beta^* + \varepsilon_i$, $\mathbb{E}[\varepsilon_i] = 0$ and $\mathrm{var}[\varepsilon_i] = \sigma^2$. Note that in this section, we replace the notation of $\theta$ with $\beta$ to present the parameters of linear regression models.

**Learning Dynamics and Implicit Regularization Effects**   The continuous-time diffusion processes for SGD algorithms with and without unbiased label noises are as follow

$$\mathbf{d}\beta^{\mathrm{ULN}}(t) = -\frac{1}{N} \sum_{i=1}^{N} x_i(x_i^\top \beta^{\mathrm{ULN}}(t) - x_i^\top \beta^*)\mathbf{d}t + \sqrt{\frac{\eta}{B}} \Sigma_N^{\mathrm{SGD}}(\beta^{\mathrm{ULN}}(t))^{1/2}\mathbf{d}W_1(t)$$

$$+ \sqrt{\frac{\eta}{B}} \Sigma_N^{\mathrm{ULN}}(\beta^{\mathrm{ULN}}(t))^{1/2}\mathbf{d}W_2(t) \quad (18)$$

$$\mathbf{d}\beta^{\mathrm{LNL}}(t) = -\frac{1}{N} \sum_{i=1}^{N} x_i(x_i^\top \beta^{\mathrm{LNL}}(t) - x_i^\top \beta^*)\mathbf{d}t + \sqrt{\frac{\eta}{B}} \Sigma_N^{\mathrm{SGD}}(\beta^{\mathrm{LNL}}(t))^{1/2}\mathbf{d}W(t)$$

where $\beta^{\mathrm{ULN}}(t)$ and $\beta^{\mathrm{LNL}}(t)$ refer to the SGD dynamics for OLS under Unbiased Label Noises and Label NoiseLess settings. We then denote the sample covariance matrix of $N$ samples as $\bar{\Sigma}_N = \frac{1}{N} \sum_{i=1}^{N} x_i x_i^\top$. Matrices $\Sigma^{\mathrm{SGD}}(\beta(t))$ and $\Sigma^{\mathrm{ULN}}(\beta(t))$ in this case are defined as

$$\Sigma_N^{\mathrm{SGD}}(\beta) = \frac{1}{N} \sum_{i=1}^{N} \left(x_i x_i^\top \beta - \bar{\Sigma}_N \beta\right)\left(x_i x_i^\top \beta - \bar{\Sigma}_N \beta\right)^\top \text{ and } \Sigma_N^{\mathrm{ULN}}(\beta) = \sigma^2 \bar{\Sigma}_N , \quad (19)$$

which are both time-homogeneous. Compared to $\beta^{\mathrm{LNL}}(t)$, the dynamics $\beta^{\mathrm{ULN}}(t)$ incorporates an additional noise term $\sqrt{\frac{\eta}{B}} \Sigma_N^{\mathrm{ULN}}(\beta^{\mathrm{ULN}}(t))^{1/2}\mathbf{d}W_2(t)$ which affects the dynamics.

**Proposition 2 (Implicit Regularization on OLS)**   *In this way, we could approximate the implicit regularizer of SGD with the random label noises for OLS through discretization such as,*

$$\sqrt{\frac{\eta}{B}} \Sigma_N^{\mathrm{ULN}}(\beta^{\mathrm{ULN}}(t))^{1/2}\mathbf{d}W_2(t) \Rightarrow \xi_k^{\mathrm{ULN}}(\beta) \approx \sqrt{\frac{\eta\sigma^2}{B}} \left(\bar{\Sigma}_N\right)^{1/2} z_k, \text{ and } z_k \sim \mathcal{N}(\mathbf{0}_d, \mathbf{I}_d), \quad (20)$$

*which is independent with $\beta$ and $k$ (the time). According to (Berthier et al., 2020), SGD for noiseless linear regression would asymptotically converge to the optimal solution $\beta^*$. With an additional noise term $\xi_k^{\mathrm{ULN}}(\beta)$ and the single optima $\beta^*$ (for both noisy and noiseless losses), we could conclude that when $k \to \infty$, SGD with unbiased random label noises would converge to a distribution centered at $\beta^*$. The distribution would tend to be a Gaussian distribution when $\sigma^2$ is significant (we could not ignore the effects of stochastic gradient noise terms of noiseless loss to the overall distribution), as the term $\sqrt{\sigma^2\eta/B}\bar{\Sigma}_N^{1/2}\mathbf{d}W(t)$ corresponds to a Gaussian distribution. The span and shape of the distribution are controlled by $\sigma^2$ and $\bar{\Sigma}_N$ when $\eta$ and $B$ are constant.*

**Numerical Validation**   To validate **Proposition 2**, we carry out numerical evaluation using synthesize data to simply visualize the dynamics over iteration of SGD algorithms with label-noisy OLS and label-noiseless OLS. In our experiments, we use 100 random samples realized from a 2-dimension Gaussian distribution $X_i \sim \mathcal{N}(\mathbf{0}, \Sigma_{1,2})$ for $1 \leq i \leq 100$, where $\Sigma_{1,2}$ is an symmetric covariance matrix controlling the random sample generation. To add the noises to the labels, we first drawn 100 copies of random noises from the normal distribution with the given variance $\varepsilon_i \sim \mathcal{N}(0, \sigma^2)$, then we setup the OLS problem with $(X_i, \tilde{Y}_i)$ pairs using $\tilde{Y}_i = X_i^\top \beta^* + \varepsilon_i$ and $\beta^* = [1, 1]^\top$ and various settings of $\sigma^2$ and $\Sigma_{1,2}$. We setup the SGD algorithms with the fixed learning rate $\eta = 0.01$, and bath size $B = 5$, with the total number of iterations $K = 1,000,000$ to visualize the complete paths.

Figure 1 presents the results of numerical validations. In Figure 1(a)–(d), we gradually increases the variances of label noises $\sigma^2$ from 0.25 to 2.0, where we can observe (1) SGD over label-noiseless OLS converges to the optimal solution $\beta^* = [1.0, 1.0]^\top$ in a fast manner, (2) SGD over OLS with unbiased random label noises would asymptotically converge to a distribution centered at the optimal point, and (3) when $\sigma^2$ increases, the span of the converging distribution becomes larger. In Figure 1(e)–(h), we use four settings of $\Sigma_{1,2}$, where we can see (4) no matter how $\Sigma_{1,2}$ is set for OLS problems, the SGD with unbiased random label noises would asymptotically converge to a distribution centered at the optimal point. Compared the results in (e) with(f), we can find that, when the trace of $\Sigma_{1,2}$ increases, the span of converging distributions would increases. Furthermore, (5) the shapes of converging distributions depend on $\Sigma_{1,2}$. In Figure 1(g), when we place the principal component of $\Sigma_{1,2}$ onto the vertical axis (i.e., $\Sigma^{\mathrm{Ver}} = [[10, 0]^\top, [0, 100]^\top]$), the distribution lays on

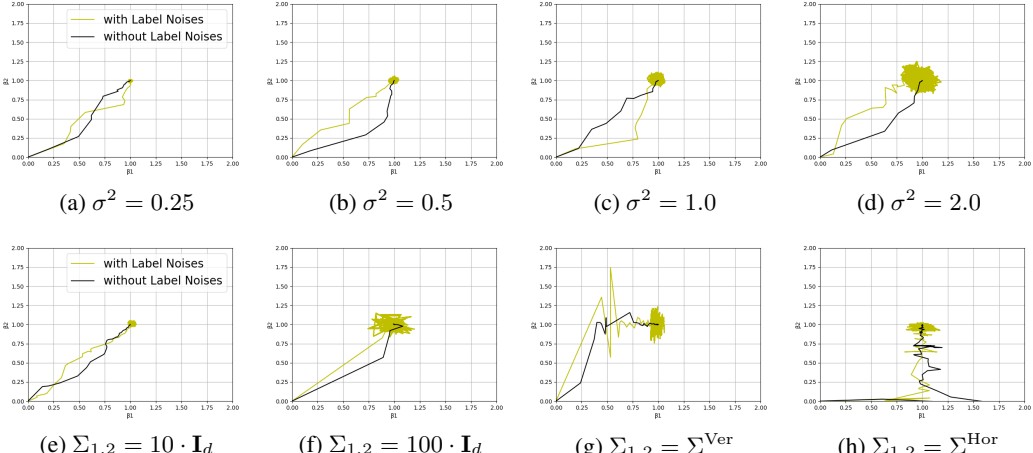

(a) $\sigma^2 = 0.25$     (b) $\sigma^2 = 0.5$     (c) $\sigma^2 = 1.0$     (d) $\sigma^2 = 2.0$

(e) $\Sigma_{1,2} = 10 \cdot \mathbf{I}_d$    (f) $\Sigma_{1,2} = 100 \cdot \mathbf{I}_d$    (g) $\Sigma_{1,2} = \Sigma^{\mathrm{Ver}}$    (h) $\Sigma_{1,2} = \Sigma^{\mathrm{Hor}}$

Figure 1: Trajectories of SGD over OLS with and without Unbiased Random Label Noises using various $\tilde{\sigma}^2$ and $\Sigma_{1,2}$ settings for (noisy) random data generation. For Figures (a)–(d), the experiments are setup with a fixed $\Sigma_{1,2} = [[20,0]^\top, [0,20]^\top]$ and varying $\tilde{\sigma}^2$. For Figures (e)–(h), the experiments are setup with a fixed $\tilde{\sigma}^2 = 0.5$ and varying $\Sigma_{1,2}$, where we set $\Sigma^{\mathrm{Ver}} = [[10,0]^\top, [0,100]^\top]$ and $\Sigma^{\mathrm{Hor}} = [[100,0]^\top, [0,10]^\top]$ to shape the converging distributions.

the vertical axis principally. Figure 1(h) demonstrates the opposite layout of the distribution, when we set $\Sigma^{\mathrm{Hor}} = [[100,0]^\top, [0,10]^\top]$ as $\Sigma_{1,2}$. The scale and shape of the converging distribution backups our theoretical investigation in Eq 20.

Note that the unbiased random label noises are added to the labels prior to the learning procedure. In this setting, it is the mini-batch sampler of SGD that "dynamizes" the noises and influences the dynamics of SGD through forming the implicit regularizer.

## 5   IMPLICIT REGULARIZATION EFFECTS TO DEEP NEURAL NETWORKS

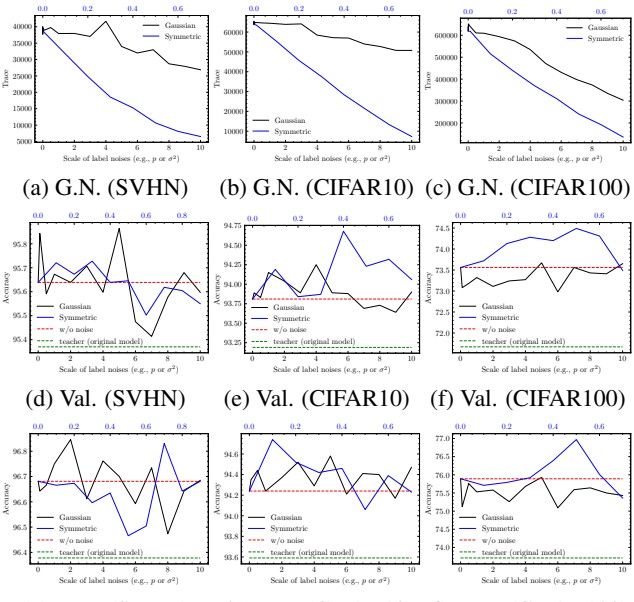

(a) G.N. (SVHN)   (b) G.N. (CIFAR10)   (c) G.N. (CIFAR100)

(d) Val. (SVHN)   (e) Val. (CIFAR10)   (f) Val. (CIFAR100)

(g) Test. (SVHN)   (h) Test. (CIFAR10)   (i) Test. (CIFAR100)

Figure 2: Gradient Norms, Validation Accuracy, and Testing Accuracy in Noisy Self-Distillation using ResNet-56 with varying scale of label noises (e.g., $p$ and $\sigma^2$).

Given a well-trained model, Noisy Self-Distillation algorithms (Zhang et al., 2019a; Xu et al., 2020; Kim et al., 2020; Xie et al., 2020) intend to further improve the performance of a model through learning from the "soft label" outputs (i.e., logits) of the model (as the teacher). Furthermore, some practices found that the self-distillation could be further improved through incorporating certain randomness and stochasticity in the training procedure so as to obtain better generalization performance (Xie et al., 2020; Kim et al., 2020). In this work, we study the way that directly adds random label noises to the logit outputs of the pre-trained model so as to improve the self-distillation (Han et al., 2018). More specifically, we study two well-known strategies for additive noises as follow.

*(1) Gaussian Noises.* Given a pre-trained model with $L$-dimensional logit output, for every iteration of self-distillation, this simple method that draw random vectors

from a $L$-dimensional Gaussian distribution $\mathcal{N}(\mathbf{0}_L, \sigma^2 \mathbf{I}_L)$, adds the vectors to the logit outputs of the model, and makes the student model learn from the noisy outputs. Note that in our analysis, we assume the output of the model is single dimension while, in self-distillation, the logit labels are with multiple dimensions. Thus, the diagonal matrix $\sigma^2 \mathbf{I}_L$ now refers to the complete form the variances and $\sigma^2$ controls the scale.

*(2) Symmetric Noises..* Basically, this strategy is derived from (Han et al., 2018) that generates noises through randomly swapping the values of logit output among the $L$ dimensions. Specifically, in every iteration of self-distillation, given a swap-probability $p$, every logit output (denoted as $y$ here) from the pre-trained model, and every dimension of logit output denoted as $y_l$, the strategy in probability $p$ swaps the logit value in the dimension that corresponds to $y_l$ with any other dimension $y_{m \neq l}$ in equal prior (i.e., in $(L-1)^{-1}$ probability). In the rest $1-p$ probability, the strategy remains the original logit output there. In this way, the new noisy label $\tilde{y}$ is with expectation $\mathbb{E}[\tilde{y}]$ as follow,

$$\mathbb{E}[\tilde{y}_l] = (1-p) \cdot y_l + \frac{p \cdot \sum_{\forall m \neq l} y_m}{L-1} \tag{21}$$

This strategy introduces explicit bias to the original logit outputs. However, when we consider the expectation $\mathbb{E}[\tilde{y}]$ as the innovative soft label, then the random noise around the new soft label is still unbiased as $\mathbb{E}[\tilde{y} - \mathbb{E}[\tilde{y}]] = 0$ for all dimensions. Note that this noise is not the symmetric noises studied for robust learning (Wang et al., 2019).

Figure 2 presents the results of above two methods with increasing scales of noises, i.e., increasing $\sigma^2$ for Gaussian noises and increasing $p$ for Symmetric noises. In Figure 2(a)–(c), we demonstrate that the gradient norms of neural networks $\frac{1}{N} \| \nabla_\theta f(x_i, \theta) \|_2^2$ decrease with growing $\sigma^2$ and $p$ for two strategies. The results backup our theoretical investigation , which means the model would be awarded high inferential stability, as the variation of neural network outputs against the potential random perturbation in parameters has been reduced by the regularization. In Figure 2(d)–(f) and (g)–(i), we plot the validation and testing accuracy of the models obtained under noisy self-distillation. The results show that (1) some of models have been improved through noisy self-distillation compared to the teacher model, (2) noisy self-distillation could obtain better performance than noiseless self-distillation, and (3) it is possible to select noisily self-distilled models using validation accuracy for better overall generalization performance. All results here are based on 200 epochs of noisy self-distillation.

## 6 DISCUSSION AND CONCLUSION

While previous studies primarily focus on the performance degradation caused by label noises or corrupted labels (Jiang et al., 2018; Li et al., 2020), we investigate the implicit regularization effects of random label noises, under mini-batch sampling settings of stochastic gradient descent (SGD). Specifically, we adopt the dynamical systems interpretation of SGD to analyze the learning procedure based on the quadratic loss with unbiased random label noises. We decompose the mini-batch stochastic gradient based on label-noisy losses into three parts in Eq. (11): (i) $\nabla L^*(\theta)$ – the true gradient of label-noiseless losses, (ii) $\xi_k^*(\theta)$ – the stochastic gradient noise caused through mini-batch sampling over the label-noiseless losses, and (iii) $\xi_k^{\mathrm{ULN}}(\theta)$ – the noise term influenced by the both random label noises and mini-batch sampling. Our research considers $\xi_k^{\mathrm{ULN}}(\theta)$ as an implicit regularizer, and finds that effects of such implicit regularizer is to lower the gradient norm of the neural networks $\frac{1}{N} \sum_{i=1}^N \| \nabla_\theta f(x_i, \theta) \|_2^2$ over the learning procedure, where the gradient norm of neural networks here characterizes the variation/stability of the neural network outputs against the random perturbation around the parameters. In summary, the new implicit regularizer $\xi_k^{\mathrm{ULN}}(\theta)$ helps SGD select a point with higher inference stability for convergence.

We carry out extensive experiments to validate our theoretical investigations. The numerical study with linear regression clearly illustrates the trajectories of SGD with and without unbiased random label noises, the observation coincides the SGD dynamics derived from our theories. Evaluation based on deep neural network shows that, in self-distillation settings, one can lower the gradient norm of neural networks, improve the inference stability of networks, and obtain better solutions, through iteratively adding noises to the outputs of teacher models. Note that we do not want to claim that the implicit regularization caused by the label noises would improve the generalization performance in this work. The experiments results well backup our theories.

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

## A    APPENDIX

### A.1    SKETCHED PROOFS IN PROPOSITION 1

To obtain Eq 16, we can use the simple vector-matrix-vector products transform that, for the random vector $v$ and symmetric matrix $A$ there has $\mathbb{E}_v[v^\top A v] = \text{trace}(A\mathbb{E}[vv^\top])$, such that

$$
\begin{aligned}
\mathbb{E}_{z_k}\|\xi_k^{\text{ULN}}(\theta)\|_2^2 &= \mathbb{E}_{z_k}\left[\xi_k^{\text{ULN}}(\theta)^\top \xi_k^{\text{ULN}}(\theta)\right] \\
&\approx \frac{\eta\sigma^2}{B}\mathbb{E}_{z_k}\left[z_k^\top\left(\frac{1}{N}\sum_{i=1}^N \nabla_\theta f(x_i,\theta)\nabla_\theta f(x_i,\theta)^\top\right)z_k\right] \\
&= \frac{\eta\sigma^2}{B}\text{trace}\left(\left(\frac{1}{N}\sum_{i=1}^N \nabla_\theta f(x_i,\theta)\nabla_\theta f(x_i,\theta)^\top\right)\mathbb{E}_{z_k}[z_k z_k^\top]\right) \\
&\quad\text{as } \mathbb{E}_{z_k}[z_k z_k^\top] = \mathbf{I}_d \; for \; z_k \sim \mathcal{N}(\mathbf{0}_d, \mathbf{I}_d) \\
&= \frac{\eta\sigma^2}{B}\text{trace}\left(\frac{1}{N}\sum_{i=1}^N \nabla_\theta f(x_i,\theta)\nabla_\theta f(x_i,\theta)^\top\right) \\
&= \frac{\eta\sigma^2}{BN}\sum_{i=1}^N \|\nabla_\theta f(x_i,\theta)\|_2^2 \\
&= \frac{\eta\sigma^2}{BN}\left\|\frac{\partial}{\partial\theta}f(x_i,\theta)\right\|_2^2 .
\end{aligned}
\tag{22}
$$

### A.2    TRAINING DETAILS FOR NOISY SELF-DISTILLATION WITH DEEP NEURAL NETWORKS

We choose the ResNet-56 (He et al., 2016), one of the most practical deep models, for conducting the experiments on three datasets: SVHN (Netzer et al., 2011), CIFAR10 and CIFAR100 (Krizhevsky et al., 2009). We follow the standard training procedure (He et al., 2016) for training a teacher model (original model). Specifically we train the model from scratch for 200 epochs and adopt the SGD optimizer with batch size 64 and momentum 0.9. The learning rate is set to 0.1 at the beginning of training and divided by 10 at $100th$ epoch and $150th$ epoch. A standard weight decay with a small regularization parameter ($10^{-4}$) is applied. As for noiseless self-distillation, we follow the standard procedure (Hinton et al., 2015) for distilling knowledge from the teacher to a student of the same network structure. The training setting is the same as training the teacher model.

For noisy self-distillation, we continue to use the training setting except the labels are noised by the two types of noises as introduced in the main text. We choose the best scale of label noises using a validation set, where we divide the original training set into a new training set (80%) and a validation set (20%). A set of $\{0.1, 0.2, ..., 0.9, 1.0\}$ is tried for the scale of symmetric noises. A set of $\{0.1, 0.5, 1.0, 2.0, 3.0, ..., 9.0, 10.0\}$ is tried for the scale of Gaussian noises. For clarity, we also present the results using all the choices of scales of label noises on test set, where the original training set is used for training.

### A.3    TRAINING PROCESS OF DEEP NEURAL NETWORKS

We show the evolution of training and test losses during the entire training procedure, and compare the settings of adding no label noises, symmetric and Gaussian noises for self-distillations. Figure 3 presents the results on the three datasets, i.e., SVHN, CIFAR10 and CIFAR100.

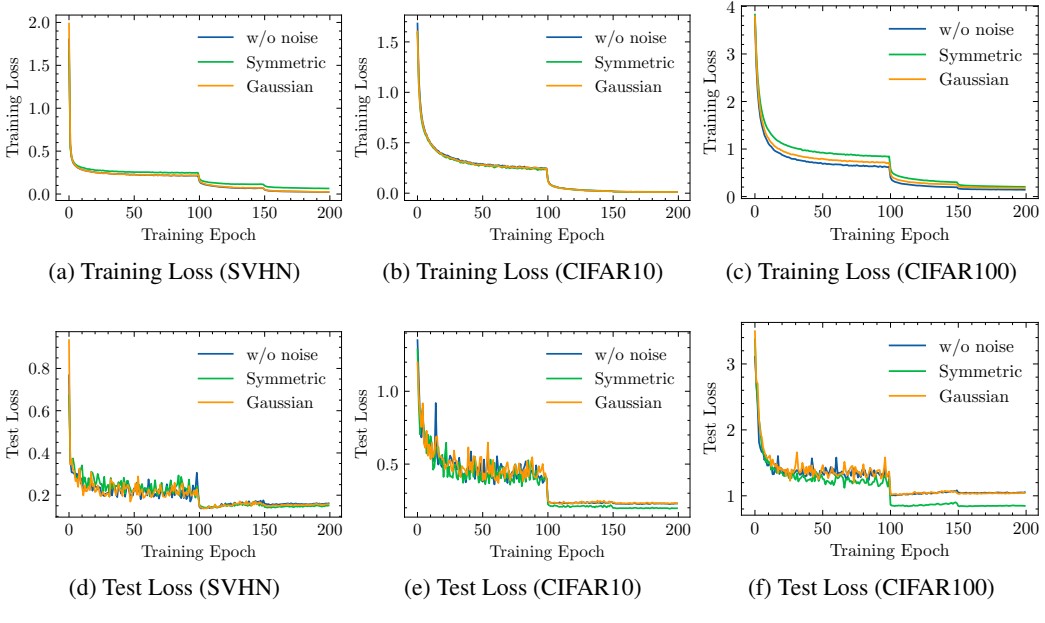

(a) Training Loss (SVHN)  (b) Training Loss (CIFAR10)  (c) Training Loss (CIFAR100)

(d) Test Loss (SVHN)  (e) Test Loss (CIFAR10)  (f) Test Loss (CIFAR100)

Figure 3

