# OpenReview forum: "Implicit Regularization Effects of Unbiased Random Label Noises with SGD"
_ICLR.cc/2021/Conference — Reject_

### Official Review · AnonReviewer4 · 2020-10-28
**overlap with previous work**

**Rating:** 3
**Confidence:** 4

**Review:**

This paper studies the implicit regularization effect from random label noise in L2 regression, and argues that it can be captured by the average squared gradient norm over samples, $\frac1N\sum_{i=1}^N \|\| \nabla_\theta f(x_i, \theta) \|\|^2$. This conclusion is achieved via analyzing the additional noise term in the update caused by the label noise, which in the SDE approximation has expected squared L2 norm proportional to $\frac1N\sum_{i=1}^N \|\| \nabla_\theta f(x_i, \theta) \|\|^2$.

The paper does not cite a very related paper:

Implicit regularization for deep neural networks driven by an Ornstein-Uhlenbeck like process. Guy Blanc, Neha Gupta, Gregory Valiant, Paul Valiant. COLT 2020

The above paper studies the same problem and obtains a similar conclusion that the implicit regularization of label noise can be captured by $\frac1N\sum_{i=1}^N \|\| \nabla_\theta f(x_i, \theta) \|\|^2$. It is proved that a solution $\theta$ which achieves zero training error is an attractive fixed point of the dynamics if and only if it is a local minimizer of the said implicit regularizer.

The high-level conclusion in the current submission is the same as Blanc et al. In addition, the current submission has a weaker theoretical argument, since it is based on the SDE approximation which may not capture SGD as the noise distribution may not be Gaussian. On the other hand, Blanc et al. rigorously prove that any attractive interpolating solution must also locally minimize the implicit regularizer. Given the overlap, the submission does not meet the bar of publication.

---

### Official Review · AnonReviewer3 · 2020-10-28
**Main result previously known**

**Rating:** 3
**Confidence:** 3

**Review:**

This paper studies the implicit regularization effect that arises from using stochastic gradient descent with label noise and squared loss. They derive the expression for the implicit regularization term and show that it favors solutions which are stable against perturbations of the parameters. This paper validates their empirical findings using SGD on linear regression task with label noise. This paper also uses their results to study the self distillation technique.

The main concern with this paper is that I think that the authors claim that they are the first ones to characterize the implicit regularization arising from SGD with label noise but this result is already known [1]. I believe that [1] works in the same setting and their implicit regularization for SGD exactly matches as derived in this paper.

I would recommend rejecting this paper because I believe that the main result presented in this paper is already known [1] and this paper has not cited or compared with that paper.

Other questions:
1) Why do the authors assume that the noise terms in equation 10 are independent when step size is small as mentioned on page 4?
2) I am not convinced by the self distillation experiments. Why do the authors consider self distillation with label noise added to the logits? Is this form of distillation considered in previous works before? Did the authors consider adding label noise to just the original labels?
3) Moreover, in the plots in figure 2, why do the curves follow this zig zag pattern with increasing noise?  Moreover, many points in the plots are actually below the red dotted line which is the accuracy without label noise. So, I am not fully convinced that label noise is actually leading to better performance.
4) It is unclear to me what is the motivation for considering these two different types of noises - gaussian and symmetric and what is the take away for these types of noises?

Some typos
1. Page 2 - extent -> extended
2. Page 2 - not clear what new convergence exactly means
3. Page 4 -  brownie -> brownian
4. Page 6 - increases -> increase

1) Implicit regularization for deep neural networks driven by an Ornstein-Uhlenbeck like process, COLT 2020

---

### Official Review · AnonReviewer2 · 2020-10-30
**Official Blind Review #2**

**Rating:** 4
**Confidence:** 4

**Review:**

The paper studies the implicit regularization effect of unbiased random label noise on the dynamics of stochastic gradient descent(SGD). More precisely, an unbiased random noise is added to the true labels and the paper aims to analyze the regularization effect of it. The paper shows that the unbiased label noise would favor convergence to points which
stabilize model outputs against perturbation of parameters.

I found the paper very obscure, rigorous justification to the hypothesis and claims are missing. For instance,
a) Below equation (12), it claims that under mild conditions, the variance of the noise vector are the given matrices. What would be the conditions? Please state it properly if it is a mild condition.
b) In the paragraph "SGD Learning Dynamics with Unbiased Random Label Noises", it assumes that terms coming from mini-batch and label noise are independent. This is not true in general, please include some positive/negative examples when the assumption holds or violates. As this is one of the key assumptions which implies the additive Brownian motion, a justification is definitely necessary showing that the assumption is valid most often in practice.
c) In the paragraph "Inference Stability/Escape and Converge", why does Prop 1 implies that the solution has low gradient norm. Intuitively, the unbiased noise increases the variance of the gradient and the algorithm would be less stable compared to noiseless version.
d) It is claimed in performance tuning that larger $\sigma$ introduces larger regularization effect with higher inference stability. If this is true, then one should use $\sigma$ as large as possible, which is clearly wrong.
e) I don't understand what is the purpose in the first experiment shown in Figure 1. It is clear that SGD ( with or without label noise) converges to a neighborhood of the solution rather than the exact minimum and the radius directly depends on the variance of the gradient. What need to be shown is the claim regarding inference stability. Moreover, one trajectory is clearly not representative as the noise $\epsilon_i$ is random. Hence experiments should be performed by drawing multiple times $\epsilon_i$, namely multiple trajectories.
Overall, there is a lack of justification regarding the claims in the paper. I won't recommend publication of the paper in the current state.

---

### Official Review · AnonReviewer1 · 2020-11-03
**Review of Implicit Regularization Effects of Unbiased Random Label Noises with SGD**

**Rating:** 2
**Confidence:** 5

**Review:**

My review is going to be short because I truly believe that this paper, in the current form, does not reach the acceptance bar for ICLR.

Indeed, the paper tries to show that SGD carries an implicit regularization along the procedure. The list of the contributions is summarized on page 2: the authors proposed a decomposition of the noise into two terms (i) a noise due to the subsampling of the gradient (ii) a noise due to the noisy labels. Neither this decomposition nor the conclusion from it is new. Worst, there are no proofs or experiments backing up there fuzzy conclusions.

Despite, the fact the story behind SGD seems understood by the authors, I see no contributions in this paper.

---

### Author Response · Authors · 2020-11-25
**Thanks for the reviews**

Many thanks for the reviews. We acknowledge that the high-level conclusions made in this manuscript already exist in the earlier work (Blanc et al. 2020), though we adopt a different model to characterize SGD and empirically evaluate the findings using deep neural networks with benchmark datasets. We regret that we failed to complete the literature review prior to the submission. In addition, the manuscript has a weaker theoretical argument than the ealier work. We acknowledge that this manuscript is not ready for publication. Thank you again for pointing all these flaws out.

---

### Decision · Program_Chairs · 2021-01-07
**Final Decision**

**Decision:**

Reject

**Comment:**

The reviewers pointed out that the claims made in this submission have already appeared (in even stronger forms) before, to which the authors seem to agree. Therefore, this submission is not ready for publication in its current form.